# Adapting an Ergosterol Extraction Method with Marine Yeasts for the Quantification of Oceanic Fungal Biomass

**DOI:** 10.3390/jof7090690

**Published:** 2021-08-26

**Authors:** Katherine Salazar Alekseyeva, Barbara Mähnert, Franz Berthiller, Eva Breyer, Gerhard J. Herndl, Federico Baltar

**Affiliations:** 1Department of Functional and Evolutionary Ecology, Microbial Oceanography Working Group, University of Vienna, 1030 Vienna, Austria; barbara.maehnert@univie.ac.at (B.M.); eva.breyer@univie.ac.at (E.B.); gerhard.hernld@univie.ac.at (G.J.H.); 2Department of Agrobiotechnology (IFA-Tulln), University of Natural Resources and Life Sciences, Vienna (BOKU), 3430 Tulln, Austria; franz.berthiller@boku.ac.at; 3Department of Marine Microbiology and Biogeochemistry, Royal Netherlands Institute for Sea Research (NIOZ), University of Utrecht, 1790 AB Den Burg, Texel, The Netherlands

**Keywords:** marine fungi, chloroform-methanol extraction, HPLC-UV, LC-MS/MS, ergosterol, pelagic fungal biomass

## Abstract

Ergosterol has traditionally been used as a proxy to estimate fungal biomass as it is almost exclusively found in fungal lipid membranes. Ergosterol determination has been mostly used for fungal samples from terrestrial, freshwater, salt marsh- and mangrove-dominated environments or to describe fungal degradation of plant matter. In the open ocean, however, the expected concentrations of ergosterol are orders of magnitude lower than in terrestrial or macrophyte-dominated coastal systems. Consequently, the fungal biomass in the open ocean remains largely unknown. Recent evidence based on microscopy and -omics techniques suggests, however, that fungi contribute substantially to the microbial biomass in the oceanic water column, highlighting the need to accurately determine fungal biomass in the open ocean. We performed ergosterol extractions of an oceanic fungal isolate (*Rhodotorula sphaerocarpa*) with biomass concentrations varying over nine orders of magnitude. While after the initial chloroform-methanol extraction ~87% of the ergosterol was recovered, a second extraction recovered an additional ~10%. Testing this extraction method on samples collected from the open Atlantic Ocean, we successfully determined ergosterol concentrations as low as 0.12 pM. Thus, this highly sensitive method is well suited for measuring fungal biomass from open ocean waters, including deep-sea environments.

## 1. Introduction

In marine environments, heterotrophic microorganisms are the main drivers of biogeochemical cycles due to their orders of magnitude higher biomass-specific metabolic rates than metazoans. The main heterotrophic organisms in terms of biomass are bacteria, followed by their grazers [1], the hetero-, and mixotrophic protists [2]. A recent study on deep-sea marine snow in the North Atlantic showed that although bacteria dominated in terms of abundance, fungi dominated the marine snow-associated microbial community in terms of biomass [3]. However, in contrast to terrestrial systems [4,5], freshwater [6,7,8], and saltmarsh studies [9], marine pelagic fungi have been poorly studied. It has been assumed that fungi only play a trivial role in the ocean, so their role in the ecology and biogeochemistry of the ocean remains largely enigmatic [10]. A metagenomic study, however, suggests that fungi are potentially involved in many biogeochemical cycles in the ocean [11]. Moreover, a recent study supports that fungi contribute to the cycling of carbohydrates in the ocean [12]. Nonetheless, most of the recent discoveries on pelagic fungal ecology are based on molecular methods that inform about the relative abundance and genetic potential and expression; however, the lack of an adequate methodology to reliably determine other basic ecological parameters such as biomass and production precludes a deeper understanding of the ecological role of fungi in the ocean.

For terrestrial and aquatic environments, the concentration of ergosterol has been used as a proxy for fungal biomass [4,5,13,14,15]. For the majority of eukaryotic organisms, sterols are essential lipids serving as structural support and precursors for hormones [16]. Ergosterol is an essential component of membranes in fungi [17,18], and its concentration is species-specific and dependent on the physiological state [8]. A property that makes ergosterol especially well-suited as a proxy for fungal biomass is that it contains a 5,7-double bond, which is rare among other sterols [9]. Additionally, it represents more than 80% of sterols in fungal strains [19]. Not all fungi are capable to synthesize ergosterol such as many fungi belonging to the division Chytridiomycota [20]. Additionally, some non-fungal organisms such as the green algae *Chlorella vulgaris* and some Protozoa can synthesize ergosterol [21,22]. However, since the division Chytridiomycota is usually either not present or a low contributor to pelagic fungal communities, and only attains relevant biomass sporadically [23], the determination of fungal biomass via ergosterol quantification appears to be suitable.

The extraction of ergosterol from a sample is the first step in the quantification of fungal biomass. The most commonly used extraction method for ergosterol is the reflux extraction [9,14] which has been used for saltmarsh material, and also the chloroform-methanol extraction [24] applied to fish tissue samples. It is noteworthy that these methods were developed and applied for environments with considerably higher fungal biomass than in the oligotrophic open ocean. Thus, it is unknown whether the currently applied ergosterol extraction and detection methods are sufficiently sensitive to determine pelagic fungal biomass.

We tested the chloroform-methanol extraction using an oceanic fungal isolate at biomass concentrations varying over nine orders of magnitude. We tested and fine-tuned the method to determine low concentrations of fungal biomass using HPLC-UV and LC-MS/MS. Finally, we applied this method to samples collected from the Atlantic Ocean throughout the water column down to bathypelagic waters.

## 2. Materials and Methods

### 2.1. Testing a Method to Allow Detecting Low Concentrations of Ergosterol Using a Cultured Fungal Strain

#### 2.1.1. Fungal Culture and Dilutions Preparation

*Rhodotorula sphaerocarpa* (HB 738), a fungus originally isolated from coastal Antarctic waters close to Marguerite Bay, was obtained from the Austrian Center of Biological Resources (ACBR). Ten fungal concentrations were prepared through serial dilution to identify the relationship between the fungal biomass and ergosterol concentration. For this, an initial amount of *Rhodotorula sphaerocarpa* cultured on yeast malt extract Agar [25,26] was diluted in 100 mL of artificial seawater (30 g/L sea salts S9883 Sigma-Aldrich) to obtain an OD_660_ ≈ 1 (i.e., 1×). The optical density (OD) was measured with a UV-1800 Shimadzu spectrophotometer. Thereafter, 100 mL of the initial fungal biomass were added to 100 mL of artificial seawater. This process was repeated sequentially to obtain dilutions of 2×, 4×, 8×, 16×, 32×, 64×, 125×, 250×, 500×, and 1000×, all in triplicate. Thirty mL from each triplicate were gently vacuum-filtered onto combusted (450 °C; 6 h) 25 mm diameter Whatman GF/F filters (WHA1825047 Sigma-Aldrich). Finally, filters were wrapped in a combusted aluminum foil and stored at −20 °C until further processing.

#### 2.1.2. Ergosterol Extraction

The chloroform-methanol method to extract ergosterol was used as described by Bligh and Dyer [24] with minor modifications. Briefly, one filter was gently shaken in a glass scintillation vial with 3 mL of 2:1 chloroform:methanol (*v*/*v*) in the dark at room temperature for 24 h. Thereafter, the extract was transferred to a 15 mL polypropylene tube, and 0.660 mL of Milli-Q water were added, thoroughly mixed, and centrifuged at 3200× *g* at room temperature for 3 min. To quantify the ergosterol potentially lost in the discarded extraction phase, the upper phase (usually discarded according to the standard protocol) was transferred to an amber glass vial. Afterwards, the lower phase was transferred to another amber glass vial and evaporated to dryness under a fume hood. Finally, the extracts were resuspended in 0.50 mL of methanol and stored at −20 °C until analysis.

The effect of repeated extractions of the same filter using the chloroform-methanol method was also tested. Each filter containing the various dilutions of *R. sphaerocarpa* was extracted three times using the chloroform-methanol approach as described above. The lower phase of each extraction was evaporated separately to dryness under a fume hood and the extracts were resuspended in 0.50 mL of methanol and stored at −20 °C until HPLC analysis.

#### 2.1.3. Determining the Extraction Efficiency of Ergosterol

To determine the extraction efficiency of the chloroform-methanol method, 7.5 mL of the biomass of OD_660_ ≈ 1 were added to 472.5 mL of artificial seawater to achieve the dilution 64×. Thirty mL were gently vacuum-filtered onto combusted (450 °C; 6 h) 25 mm diameter Whatman GF/F filters (WHA1825047 Sigma-Aldrich). In total, 12 filters were prepared in this way and each filter was wrapped in a combusted aluminum foil and stored at −20 °C until further processing. Before the extraction, 0.5 mL of ergosterol standards in methanol (1400 nM, 700 nM, and 350 nM final concentration; HPLC grade, Sigma-Aldrich) were added to each filter (containing the biomass) in triplicate, and no ergosterol standard was added to three filters. Subsequently, the filters were placed in the oven at 40 °C overnight. Finally, they were extracted using chloroform-methanol extraction as described above, and the extracts were analyzed by HPLC analysis.

#### 2.1.4. Ergosterol Quantification by High-Performance Liquid Chromatography (HPLC-UV) Analysis

Ergosterol concentrations were quantified by applying HPLC according to the protocol of Gulis and Bärlocher [27]. An Agilent 1260 Infinity Bioinert HPLC System equipped with an autosampler was used together with a Zorbax StableBond Aq Analytical Guard Column (4.6 × 12.5 mm; 5 µm particle size; 80 Å pore size, Agilent Technologies, Santa Clara, CA, USA) and a Zorbax Eclipse AAA Rapid resolution column (4.6 × 150 mm; 3.5 µm particle size, 80 Å pore size, Agilent Technologies). For detecting ergosterol, the wavelength of the UV detector was set at 282 nm. The mobile phase consisted of 100% methanol (HPLC grade, Sigma-Aldrich) with a flow rate of 0.8 mL/min. The injection volume was 400 µL and the samples were run for 20 min in total, including 5 min of purging at a column temperature of 25 °C. Ergosterol standards (HPLC grade, Sigma-Aldrich) were prepared in a range from 500 µM to 30 nM in methanol and run together in the same sequence batch with the fungal samples. Furthermore, the temperature of the autosampler was kept at 10 °C to protect them from potential degradation. Peak areas were used for calculating ergosterol concentrations in samples.

#### 2.1.5. Limit of Detection (LOD) and Limit of Quantitation (LOQ) of Ergosterol by HPLC Analysis

The lowest ergosterol concentration that can be reliably measured by HPLC was determined according to the protocol of Wenzl et al. [28]. Six ergosterol standards (HPLC grade, Sigma-Aldrich) were prepared in five replicates ranging from 10 µM to 30 nM in methanol. Linear calibrations were performed and LOD and LOQ concentrations were calculated.

### 2.2. Measuring Ergosterol Concentrations in Samples Collected throughout the Water Column of the Atlantic Ocean

#### 2.2.1. Sampling

Samples were collected from the water column of the Atlantic Ocean in March-April 2019 during the Poseidon expedition on board the R.V. *Sarmiento de Gamboa* from Punta Arenas, Chile to Santa Cruz de Tenerife, Spain. Seawater was collected with a CTD rosette sampler containing 12 L Niskin bottles from three different depths corresponding to epipelagic (5 m), mesopelagic (950 m), and bathypelagic (4000 m) waters. Ten liters of seawater were vacuum-filtered onto combusted 47 mm diameter Whatman GF/F filters (WHA1825047 Sigma-Aldrich). Filters were wrapped in combusted aluminum foil and stored at −20 °C until analyses.

#### 2.2.2. Ergosterol Extraction and Concentration

The ergosterol extraction of the samples collected in the Atlantic Ocean was performed with the chloroform-methanol method as mentioned above with minor modifications. Each filter was extracted twice, so the lower phases from extraction 1 and 2 were transferred to the same amber glass vial and evaporated to dryness under a fume hood. Finally, the dried extract was resuspended in 0.1 mL of methanol and stored at −20 °C until analysis.

#### 2.2.3. Ergosterol Quantification via Liquid Chromatography–Mass Spectrometry/Mass Spectrometry (LC-MS/MS) Analysis

To ensure the identity of the analyte applying the HPLC-UV method and to compare quantification, LC-MS/MS analysis was also performed. Ergosterol concentrations were quantified with LC-MS/MS similar to the protocol of Ory et al. [29]. Briefly, an Agilent (Waldbronn, Germany) 1290 UHPLC System was used together with a Sciex (Framingham, MA, USA) QTrap 6500+ mass spectrometer and a Phenomenex (Aschaffenburg, Germany) Gemini column (150 × 4.6 mm, 5 µm particle size). Isocratic elution at 35 °C was performed with a mobile phase consisting of acidified acetonitrile (HPLC grade, Sigma-Aldrich) with 0.1% formic acid at a flow rate of 2.5 mL/min. The ergosterol standards (HPLC grade, Sigma-Aldrich) were prepared in a range from 1 nM to 100 µM in methanol and an injection volume of 20 µL was used. The retention time was 3.78 min, and the eluent was sent to the atmospheric pressure chemical ionization (APCI) source between 3.0 to 4.5 min.

For the ionization of ergosterol in positive ion mode, a heated nebulizer APCI probe was used in the Turbo V source with the following settings: curtain gas 30 psi, nebulizer gas 40 psi, temperature 500 °C, nebulizer current 3 µA. The mass spectrometer was operated in selected reaction monitoring mode using both the [M+H-H_2_O]^+^ (declustering potential 100 V) and the unusual [M+H-2H_2_]^+^ (declustering potential 50 V) ions as precursors. The following six transitions were monitored for 50 ms each (Q1 mass > Q3 mass (collision energy, CE)): 379.3 > 295.2 (20 eV), 379.3 > 159.1 (30 eV), 379.3 > 145.1 (35 eV), 379.3 > 69.0 (55 eV), 393.3 > 268.2 (30 eV), 393.3 > 173.1 (35 eV). Analyst version 1.6.3 was used for data acquisition and evaluation.

### 2.3. Correlation between HPLC-UV and LC-MS/MS

For the correlation between the two detection methods, we performed a growth experiment. *Sakaguchia dacryoidea* (HB 877), a fungus originally isolated from the Antarctic waters, was obtained from the Austrian Center of Biological Resources (ACBR). An initial amount of *Sakaguchia dacryoidea* cultured on yeast malt extract Agar [25,26] was diluted in 10 mL of artificial seawater (30 g/L sea salts S9883 Sigma-Aldrich) to obtain an OD_660_ ≈ 1. Afterwards, these 10 mL were added to 1000 mL of liquid media containing: 2 g glucose, 2 g peptone, 2 g yeast extract, 2 g malt extract, 30 g artificial sea salts, and 0.5 g chloramphenicol. The liquid cultures were divided into ten bottles, each containing 100 mL of liquid media + fungi and incubated at 140 rpm (Aggro Lab, Ski 4) and at room temperature. The cultures were sampled in the lag, exponential, and stationary phase as determined by their optical density. Thirty milliliters from each triplicate were gently vacuum-filtered onto combusted (450 °C; 6 h) 47 mm diameter Whatman GF/F filters (WHA1825047 Sigma-Aldrich). Filters were wrapped in a combusted aluminum foil and stored at −20 °C until further processing. Finally, the filter was extracted using the chloroform-methanol approach as described above and the extracts were analyzed via HPLC-UV and LC-MS/MS analysis.

## 3. Results and Discussion

Ergosterol is the most abundant fungal sterol [30] and it is almost exclusively produced by fungi [13]. Hence, it has been used to determine fungal biomass in terrestrial and aquatic environments [19]. However, quantifying ergosterol in open ocean waters and the deep sea requires highly sensitive methods due to the expected low fungal abundance in these environments. Accordingly, we extracted and concentrated ergosterol, and compared two detection methods.

The chloroform-methanol approach allows ergosterol extraction and purification in a single operation. This method is also less time-consuming and no special equipment is required such as a rotavapor as compared to the reflux method [9,14]. According to Bligh and Dyer [24], the chloroform-methanol extraction method produces a biphasic system where the chloroform layer contains the lipids and the aqueous layer contains the non-lipids. Nonetheless, we determined the ergosterol concentration in both the lower and upper phase to test whether ergosterol is carried over and discarded with the upper phase. Most of the ergosterol concentration was recovered in the lower phase (chloroform layer), irrespective of the fungal dilution (Figure 1). Approximately 98% of the extracted ergosterol was contained in the lower phase, whereas less than 2% was found in the upper phase. Ergosterol has a logP value of 8.86, which reflects the molecule affinity to the organic portion, in this case, chloroform. Moreover, this extraction allowed a recovery of ~86% of ergosterol standards. For a toluene extraction, a similar value (90%) was obtained by Verma et al. [18].

The LOD for HPLC-UV was 37.2 nM, while the LOQ was 122.8 nM. As a result, we were able to determine ergosterol concentrations in 7 out of 9 fungal dilutions (4× to 250×) (Figure 2A and Figure 3). Due to high biomass, 30 mL of the dilution 2× could not be filtered onto a 25 mm diameter Whatman GF/F filter, so it was not measured. The ergosterol concentration was linearly related to fungal biomass using *Rhodotorula sphaerocarpa* (Figure 4). Consequently, the chloroform-methanol extraction resulted in a good relationship between ergosterol concentration and fungal biomass (R^2^= 0.9715).

We also tested the significance of a second and third extraction (Figure 2B,C). In the study of Bligh and Dyer [24] and Roose and Smedes [31], a second extraction yielded an additional lipid recovery of 6% and 8–10%, respectively. We found that a second extraction accounts for 5–16% of the total extracted ergosterol in the sample. For fungal dilutions 4×, 8×, 16×, 32×, 64×, and 125×, a second extraction allowed an additional recovery of 7.6%, 12.6%, 4.9%, 15.1%, 10.3%, and 15.5%, respectively. A third extraction led to an additional 1% of the total extracted ergosterol. Therefore, we recommend performing a second extraction, particularly when very low fungal biomass is expected as in oceanic samples.

The two detections, HPLC-UV and LCMS-MS were in good agreement (R^2^ = 0.9985) (Figure 5). Only in the lag phase, where the lowest ergosterol concentrations were detected in the culture, both methods differed remarkably with 380.2 ± 59.0 nM for HPLC-UV and 97.6 ± 57.6 nM LC-MS/MS. In open ocean waters and the deep sea, highly sensitive methods are required to quantify ergosterol as a low fungal abundance is expected in these environments. Thus, for oceanic samples collected from the Atlantic Ocean over three depths, we determined ergosterol with LC-MS/MS.

For the developed LC-APCI-MS/MS method, the observed ionization was in agreement with results obtained from Ory et al. [29], with the [M+H-H_2_O]^+^ and the [M+H-2H_2_]^+^ ions being the most intense. An LOD of 1.5 nM (S/N = 3/1) and an LOQ of 5.0 nM (2 ng/mL) at S/N = 10/1 could be obtained, thus being about 25 times more sensitive than with the HPLC-UV method. A chromatogram of an ergosterol standard (1 µM) with all six SRM transitions is shown in Figure 6. We were able to successfully concentrate ergosterol and detect it in the pM range (Table 1). Marine samples were concentrated 10,000-fold (10 L water filtered and finally taken up in 0.1 mL); thus, the LOQ in seawater was about 0.05 pM (0.02 pg/mL). The highest concentration of ergosterol was 0.31 pM corresponding to epipelagic (5 m), whereas the lowest concentration was below the limit of detection corresponding to bathypelagic (4000 m). In the mesopelagic (950 m), the concentration was 0.12 pM. These numbers correspond to those of Hassett et al. [15], reporting an ergosterol concentration of 0.12 pM at a 246 m depth in the Arctic Ocean.

Overall, we successfully adapted a method originally developed for ergosterol extraction from fish tissue [24] to pelagic fungi. Based on our findings, we propose an ergosterol extraction method (detailed in Figure 7) and quantification method specially adapted for low fungal concentrations. We found three crucial steps in the determination of ergosterol which are (1) a re-extraction of the filter, (2) the volume of the extract, and (3) the filtration volume of the water sample. A second extraction of the filter accounts for an additional ~10% of the ergosterol recovered. Additionally, the final volume of the extraction was adjusted to 0.5 mL for culture samples and 0.1 mL for oceanic samples, which might also be essential for the ergosterol quantification. Nonetheless, as mentioned before, it is important to consider that the ergosterol concentration can vary between species, and also that it can be influenced by different environmental parameters [6,7,8]. Thus, it is now relevant to test this method with other fungal species from oceanic origin (including hyphae like), and to consider the importance of environmental conditions on the ergosterol concentration when estimating the biomass of oceanic fungi. In spite of these few limitations, the method characterized here represents an important step for future research on the ecological role of fungi in the ocean and an adequately sensitive method for environments with low fungal biomass.

## Figures and Tables

**Figure 1 jof-07-00690-f001:**
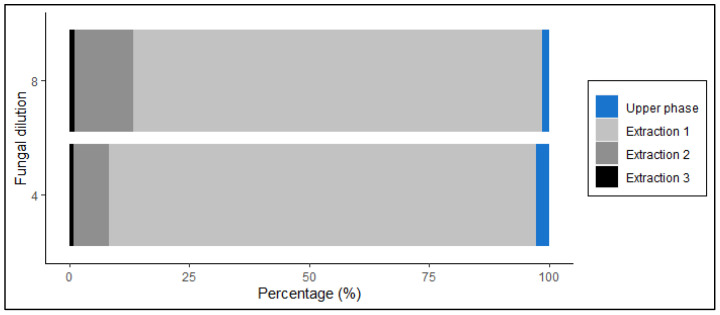
Percentages of ergosterol extracted with chloroform-methanol approach from two fungal dilutions (4×, and 8×) of the marine fungal isolate *Rhodotorula sphaerocarpa*. Extracts recovered from the upper (one extraction) and lower (three extractions) phases were analyzed.

**Figure 2 jof-07-00690-f002:**
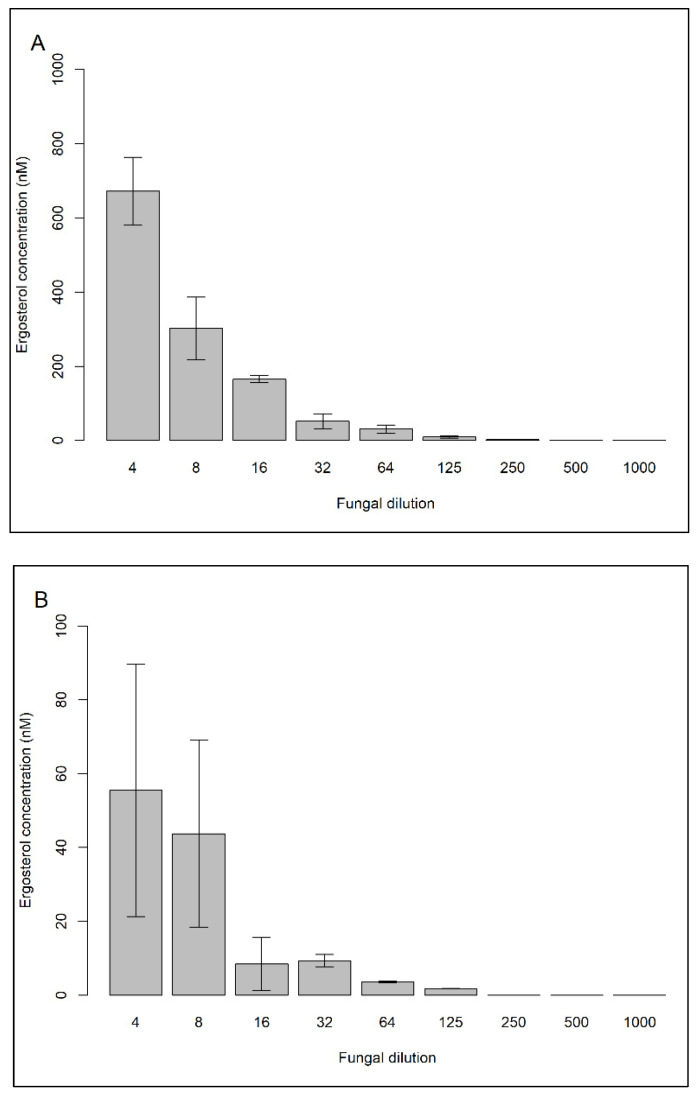
Ergosterol concentration in nM obtained with chloroform-methanol extraction from the marine fungal isolate *Rhodotorula sphaerocarpa*. Nine fungal dilutions in artificial seawater were used (4×, 8×, 16×, 32×, 64×, 125×, 250×, 500×, and 1000×). Extracts from the first (**A**), second (**B**), and third (**C**) extractions were analyzed.

**Figure 3 jof-07-00690-f003:**
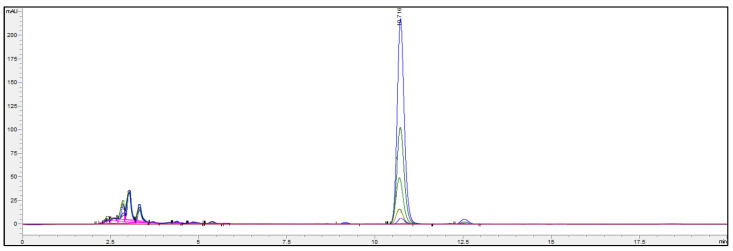
LC-UV-chromatogram of ergosterol obtained with chloroform-methanol extraction from the marine fungal isolate *Rhodotorula sphaerocarpa*. Five fungal dilutions in artificial seawater are shown (4×, 8×, 16×, 32×, and 64×), and the *x*-axis shows minutes of the run.

**Figure 4 jof-07-00690-f004:**
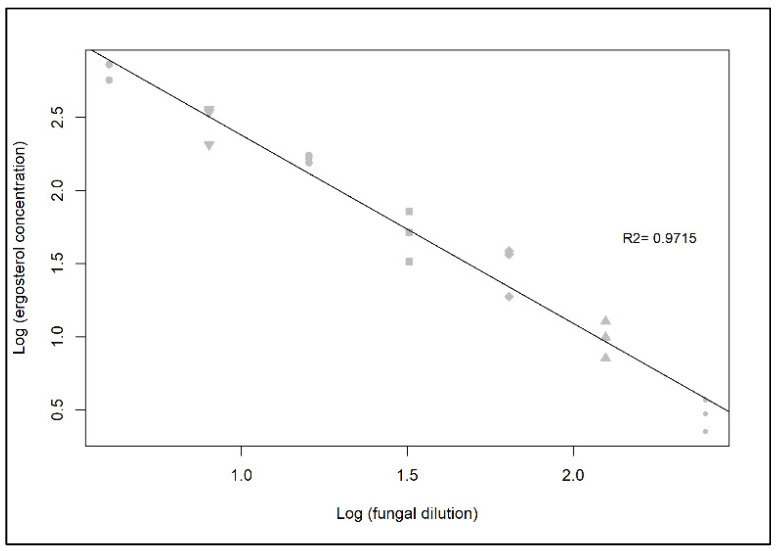
Relationship between ergosterol concentration and fungal biomass dilution.

**Figure 5 jof-07-00690-f005:**
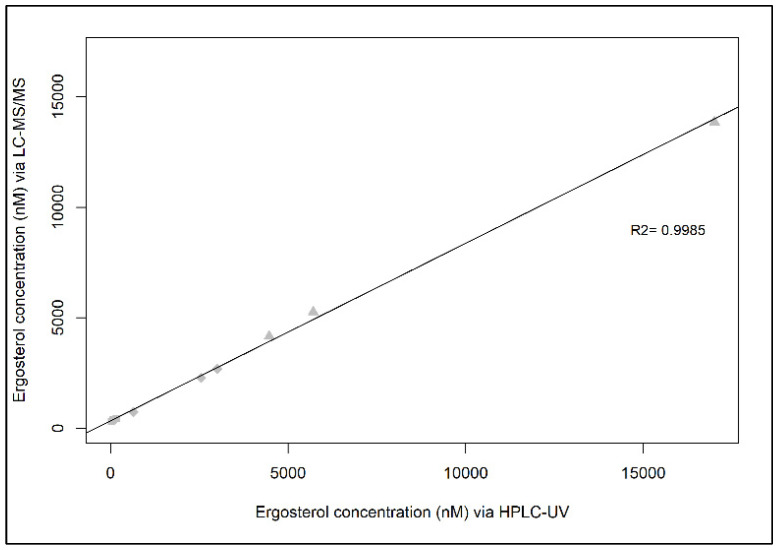
Ergosterol content in nM detected via HPLC-UV vs. LC-MS/MS from the marine fungal isolate *Sakaguchia dacryoidea* covering three growth phases (lag, exponential, and stationary).

**Figure 6 jof-07-00690-f006:**
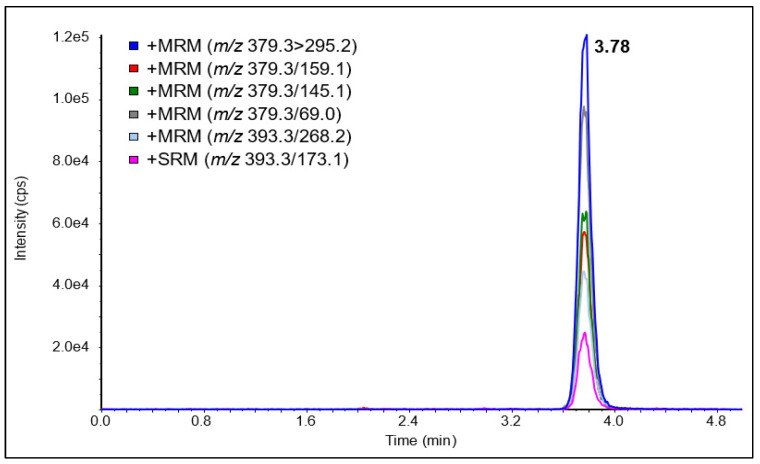
LC-APCI-MS/MS chromatogram of a 1000 nM ergosterol standard.

**Figure 7 jof-07-00690-f007:**
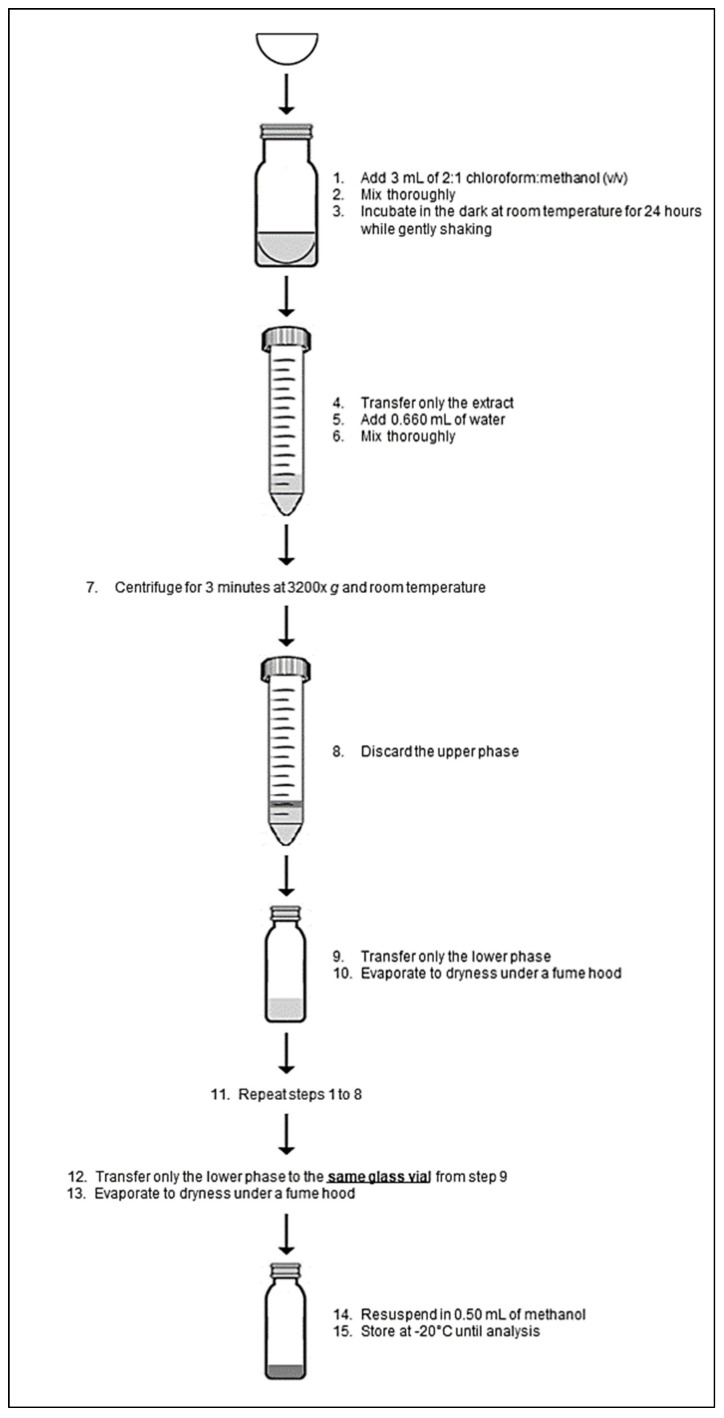
Ergosterol extraction method modified and adapted for pelagic fungi.

**Table 1 jof-07-00690-t001:** Ergosterol content in the seawater of the Atlantic Ocean obtained with the chloroform-methanol approach (Bligh and Dyer 1959).

Coordinates		Pelagic Zone	Depth (m)	Ergosterol Concentration (pM)	Ergosterol Concentration (ng/L)
Latitude	Longitude				
13°35.748	−29°42.830	Epipelagic	5	0.306	0.121
		Mesopelagic	950	0.120	0.048
		Bathypelagic	4000	0.000	0.000

## Data Availability

The raw data supporting the conclusions of this article will be made available by the authors, without undue reservation to any qualified researcher.

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
