# Peer review of "Adapting an Ergosterol Extraction Method with Marine Yeasts for the Quantification of Oceanic Fungal Biomass"

_jof, 2021, doi:10.3390/jof7090690_

Round 1

Reviewer 1 Report

Manuscript ID: jof-1329481

In the manuscript entitled “Adapting an ergosterol extraction method for the quantification of oceanic fungal biomass”, the authors developed a highly sensitive method to determined ergosterol concentrations  as a measure fungal biomass from open ocean waters. The manuscript is well written and suitable for publication in JOF with minor revisions.

  1. Error! Reference source not found (Lines 237, 281).
  2. Remove block around Figure 5.
  3. Figure 7 very difficult to read.

Author Response

Response to Reviewer 1 Comments

We acknowledge Reviewer 1 for his/her helpful comments on the paper “Adapting an ergosterol extraction method for the quantification of oceanic fungal biomass”. 

Point 1: In the manuscript entitled “Adapting an ergosterol extraction method for the quantification of oceanic fungal biomass”, the authors developed a highly sensitive method to determined ergosterol concentrations  as a measure fungal biomass from open ocean waters. The manuscript is well written and suitable for publication in JOF with minor revisions.

We appreciate the contribution of Reviewer 1 to this paper.

Point 2: Error! Reference source not found (Lines 237, 281).

This has been now corrected.

Point 3: Remove block around Figure 5.

We are not sure what block the reviewers refers to, we do not find any block around the figure, just the margin which is the same in all other figures. 

Point 4: Figure 7 very difficult to read.

We have changed the Figure 7 quality and size.

Reviewer 2 Report

The manuscript “Adapting an ergosterol extraction method for the quantification of oceanic fungal biomass” describes a modified method for the determination of ergosterol in seawater in order to assess the low fungal biomass characteristic of this ecosystem. Authors should strengthen their arguments for the purpose of their work, as molecular methods are increasingly used to detect and count the number of various microorganisms including fungi. In general, the article is not bad, adequate methods have been applied, and the results can be useful for many researchers.

There is one essential remark to the work. The authors did not provide data on the quantitative content of fungi in the marine samples that they studied. Several methods are used to estimate the biomass of microorganisms. The use of turbidity makes sense if the turbidity is related to the dry biomass of the fungus per unit volume. The article does not present such data, but extrapolation to the logarithm of dilutions was carried out. Dilutions are not quantitative and cannot be used in this case.

I have a few more minor remarks:

- some references are missing in the text;

- the list of references contains references that are not cited in the text;

-Figure 5 Ergosterol content detected….

Table title: The content of ergosterol in the seawater of the Atlantic Ocean.

Depth, concentration units are given in the table and there is no need to write this in the title. There should be one more column in this table - Fungi biomass, mg or ng mL-1 or L-1

In conclusion, it is imperative to indicate the limiting sensitivity of the method.

Author Response

Response to Reviewer 2 Comments

We acknowledge Reviewer 2 for his/her helpful comments on the paper “Adapting an ergosterol extraction method for the quantification of oceanic fungal biomass”.

Point 1: The manuscript “Adapting an ergosterol extraction method for the quantification of oceanic fungal biomass” describes a modified method for the determination of ergosterol in seawater in order to assess the low fungal biomass characteristic of this ecosystem. Authors should strengthen their arguments for the purpose of their work, as molecular methods are increasingly used to detect and count the number of various microorganisms including fungi. In general, the article is not bad, adequate methods have been applied, and the results can be useful for many researchers.

We are aware of the existence of other methods for the detection of fungi. Molecular methods are good for the determination of relative abundance of microbes but not really appropriate for accurate determination of biomass values. Nevertheless, with this adapted method, we would like to provide an option to researchers of a fast and relatively low-cost method, compared with other methods.

We have also now included a sentence on this in the manuscript; it reads: “Nonetheless, most of the recent discoveries on pelagic fungal ecology are based on molecular methods that inform about the relative abundance and genetic potential and expression, however the lack of an adequate methodology to reliably determine other basic ecological parameters such as biomass and production precludes a deeper understanding of the ecological role of fungi in the ocean.”

Point 2: There is one essential remark to the work. The authors did not provide data on the quantitative content of fungi in the marine samples that they studied. Several methods are used to estimate the biomass of microorganisms. The use of turbidity makes sense if the turbidity is related to the dry biomass of the fungus per unit volume. The article does not present such data, but extrapolation to the logarithm of dilutions was carried out. Dilutions are not quantitative and cannot be used in this case.

We used optical density (absorbance, 660nm) as an initial reference for starting our experiments. From there, we just performed serial dilutions in triplicates, which is a very common approach in chemical and biological studies. Based on our results, both the triplicates and the dilution patterns observed, indicate that there is no reason to assume that the serial dilutions did not work.

Point 3: Some references are missing in the text.

This has now been corrected.

Point 4: The list of references contains references that are not cited in the text.

This has now been corrected.

Point 5: Figure 5 Ergosterol content detected….

The change has been applied.

Point 6: Table title: The content of ergosterol in the seawater of the Atlantic Ocean.

The change has been applied.

Point 7: Depth, concentration units are given in the table and there is no need to write this in the title.

The change has been applied.

Point 8: There should be one more column in this table - Fungi biomass, mg or ng mL-1 or L-1

The concentration in ng/L has been added.

Point 9: In conclusion, it is imperative to indicate the limiting sensitivity of the method.

We appreciate the contribution of Reviewer 2 on this paper.

Round 2

Reviewer 2 Report

The authors have responded to almost all of my comments and the article can be accepted. But I strongly recommend changing the table title. Table 1 title should be “Ergosterol content in the seawater of the Atlantic Ocean obtained with the chloroform-methanol approach ”.

Author Response

Response to Reviewer 2 Comments

We acknowledge once again Reviewer 2 for his/her helpful comments on the paper “Adapting an ergosterol extraction method for the quantification of oceanic fungal biomass”.

Point 1: The authors have responded to almost all of my comments and the article can be accepted. But I strongly recommend changing the table title. Table 1 title should be “Ergosterol content in the seawater of the Atlantic Ocean obtained with the chloroform-methanol approach”.

The change in the Table 1 title has been applied.

We appreciate the contribution of Reviewer 2 on this paper.
